# Towards Scalable Coverage-Based Testing of Autonomous Vehicles

**James Tu**[1,2]   **Simon Suo**[1,2]   **Chris Zhang** [1,2]   **Kelvin Wong**[1,2]   **Raquel Urtasun**[1,2]

[1]Waabi   [2]University of Toronto

{jtu,czhang,kwong,urtasun}@waabi.ai suo@cs.toronto.edu

**Abstract:** To deploy autonomous vehicles(AVs) in the real world, developers must understand the conditions in which the system can operate safely. To do this in a scalable manner, AVs are often tested in simulation on parameterized scenarios. In this context, it's important to build a testing framework that partitions the scenario parameter space into safe, unsafe, and unknown regions [1]. Existing approaches rely on discretizing continuous parameter spaces into bins, which scales poorly to high-dimensional spaces and cannot describe regions with arbitrary shape. In this work, we introduce a problem formulation which avoids discretization — by modeling the probability of meeting safety requirements everywhere, the parameter space can be paritioned using a probability threshold. Based on our formulation, we propose GUARD as a testing framework which leverages Gaussian Processes to model probability and levelset algorithms to efficiently generate tests. Moreover, we introduce a set of novel evaluation metrics for coverage-based testing frameworks to capture the key objectives of testing. In our evaluation suite of diverse high-dimensional scenarios, GUARD significantly outperforms existing approaches. By proposing an efficient, accurate, and scalable testing framework, our work is a step towards safely deploying autonomous vehicles at scale.

**Keywords:** Testing, Coverage, Self-Driving

## 1   Introduction

Autonomous vehicles (AVs) will soon become a staple in ground transportation—interacting with billions of people everyday. At this scale, AVs can drastically reduce accidents, relieve traffic congestion, and provide mobility for those who cannot drive. In order to realize this future, developers must first ask the question: *"Is the AV safe enough to be deployed in the real world?"* To understand how safe the AV is, it's important to identify in which scenarios the AV is safe or unsafe with respect to requirements defined by safety experts [1, 2]. Towards this goal, it's important to build a testing framework which *covers* the wide range of scenarios in the AV's operational domain and classify them as safe, unsafe, or unknown.

To cover the wide range of real-world scenarios in a scalable manner, the AV industry often builds testing frameworks in simulation [3, 4, 5] where the traffic environment is fully controllable and long-tail events can be synthesized. A popular strategy to describe real world events in simulation involves designing parameterized scenarios [6]. These scenarios describe semantics of the environment (e.g. truck merging from on-ramp) and its parameters specify low-level characteristics (e.g. velocity of traffic participants). Each parameter configuration then corresponds to a concrete test which can be executed in simulation to determine if the AV complies with functional safety requirements [1]. This is typically captured as a binary pass or fail determined by regulatory demand [7]. For example, an AV could fail if it violates a safety distance threshold.

However, directly covering all variations in a scenario's parameter space can be infeasible. Continuous parameters imply infinitely many variations and testing frameworks can only execute a finite number of concrete tests which is limited by computation budget. As a result, to cover the parameter space the testing framework must leverage observed test results to estimate if the AV will pass or fail on unseen test configurations. Furthermore, the testing framework must also efficiently choose concrete tests to execute to maximize coverage and accurately estimate AV performance.

7th Conference on Robot Learning (CoRL 2023), Atlanta, USA.

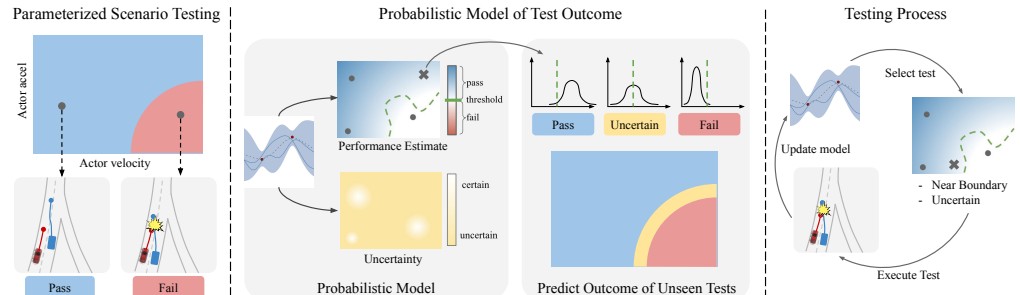

Figure 1: **Left:** Parameterized scenario testing. **Middle:** Probabilistic model estimates pass/fail outcomes in parameter space. **Right:** Testing process—sample and execute tests, then update model.

Existing approaches rely heavily on *discretization* to reduce the infinite continuous parameter space into discrete bins, making the assumption that tests in the same bin yield the same result. While effective for simple 2-dimensional scenarios [8, 9], they scale poorly to high dimensional scenarios due to exponentially growing number of bins. Alternatively, another common approach involves generating adversarial examples [10, 11, 12] to discover failures. While revealing failure cases is useful, these methods forgo covering the parameter space and cannot provide an understanding of AV performance across the parameter space.

In this work, we formulate the testing process as sequencing a finite number of tests to estimate the probability of passing or failing across the entire parameter space, thereby avoiding discretization altogether. To start, we can leverage observed test results to model the probability of passing or failing across the *entire* parameter space. Intuitively, if the AV easily passed a particular test, it can pass similar tests with high probability. Moreover, for an unseen test with little information from similar tests, the pass/fail outcome should be ambiguous. By modeling the probability everywhere, it follows that a probability threshold can partition the parameter space into 1) pass regions, 2) failure regions, and 3) uncertain regions, which is the output prescribed by standard safety guidelines [1, 2].

Based on our formulation, we propose our testing framework GUARD. Specifically, we leverage Gaussian Processes (GPs) to model the probability of passing and failing. To achieve high sample efficiency, we adopt levelset algorithms which adaptively generates tests that balance exploration (of high uncertainty regions) and exploitation (by sampling near the pass/fail boundary). Moreover, GUARD leverages online GP kernel learning, allowing it to scale to different parameterized scenarios with minimal hyperparameter tuning.

To evaluate testing frameworks, we propose a suite of novel evaluation metrics which capture the key objectives of testing: achieving high coverage, evaluating safety of the AV, and exposing failure cases for autonomy development. Under these metrics, GUARD is able to outperform existing testing frameworks across a diverse set of complex and high-dimensional driving scenarios. Moreover, we find that by avoiding discretization, our approach scales much more efficiently as the dimensionality of test scenarios grow. Finally, GUARD can also benchmark different iterations of autonomy models while identifying specific examples of scenarios where regression occurs.

## 2  Related Work

### 2.1  Coverage-Based Testing

AVs are often tested in simulation using parameterized scenarios [6, 13, 14] where it receives a pass or fail based on safety requirements [7]. Each parameter configuration corresponds to a concrete test and testing frameworks execute concrete tests to cover the parameter space. Since continuous scenario parameters imply an infinite amount of scenario variations, existing works *discretize* the parameter space into bins — assuming all pass/fail outcomes in a bin are identical. This discretization introduces error in understanding where the AV passes and fails, which grows with the bin size.

Exhaustively testing all parameter bins [14, 15] is a widely adopted approach in industry. However, this approach suffers from coarse discretization when testing high-dimensional scenarios. To increase sample efficiency, $t$-way testing [16] samples points such that for every subset of $t$ parameters, every combination is covered. This affords finer discretization but leads to inaccuracies due to assuming that tests which share a subset of $t$ parameters yield the same result. Similarly,

**Algorithm 1:** Algorithm for GUARD

---

**Input:** Threshold $\gamma$, threshold $\alpha$, budget $N$, initial budget $M$, kernel params $\boldsymbol{\kappa}_0$, update freq $K$

$G \leftarrow GP(\boldsymbol{\kappa}_0)$      `// Initialize GP`

**for** $t$ *in 1, ..., M* **do**      `// Initial exploration batch`
>   $\boldsymbol{\theta}_t \leftarrow \text{argmax}_{\boldsymbol{\theta}} \, \sigma(\boldsymbol{\theta})$
>   $G \leftarrow \texttt{UpdateGPSamples}(G, \boldsymbol{\theta}_t, f^*(\boldsymbol{\theta}_t))$

$\boldsymbol{\kappa}_M \leftarrow \text{argmax}_{\boldsymbol{\kappa}} \, P(f(\boldsymbol{\theta}_1) \dots f(\boldsymbol{\theta}_t) \,|\, \boldsymbol{\theta}_1 \dots \boldsymbol{\theta}_t; \boldsymbol{\kappa})$      `// Fit kernel on initial batch`

$G \leftarrow \texttt{UpdateGPKernel}(G, \boldsymbol{\kappa}_M)$

**for** $t$ *in M+1, ..., N* **do**      `// Levelset sampling`
>   $\boldsymbol{\theta}_t \leftarrow \text{argmax}_{\boldsymbol{\theta}} \, \beta\sigma(\boldsymbol{\theta}) - |\mu(\boldsymbol{\theta}) - \gamma|$      `// Explore or close to boundary`
>   $G \leftarrow \texttt{UpdateGPSamples}(G, \boldsymbol{\theta}_t, f^*(\boldsymbol{\theta}_t))$
>   **if** $t \equiv 0 \mod K$ **then**      `// Fit kernel every K steps`
>   >   $\boldsymbol{\kappa}_t \leftarrow \text{argmax}_{\boldsymbol{\kappa}} \, P(f(\boldsymbol{\theta}_1) \dots f(\boldsymbol{\theta}_t) \,|\, \boldsymbol{\theta}_1 \dots \boldsymbol{\theta}_t; \boldsymbol{\kappa})$
>   >   $G \leftarrow \texttt{UpdateGPKernel}(G, \boldsymbol{\kappa}_t)$

$\mathcal{P} \leftarrow \{\boldsymbol{\theta} \in \boldsymbol{\Theta} \mid \Phi(\frac{\gamma - \mu(\boldsymbol{\theta})}{\sigma(\boldsymbol{\theta})}) \geq \alpha\}$      `// Pass region`

$\mathcal{F} \leftarrow \{\boldsymbol{\theta} \in \boldsymbol{\Theta} \mid \Phi(\frac{\mu(\boldsymbol{\theta}) - \gamma}{\sigma(\boldsymbol{\theta})}) \geq \alpha\}$      `// Fail region`

$\mathcal{U} \leftarrow \boldsymbol{\Theta} \setminus \mathcal{P} \setminus \mathcal{F}$      `// Unknown region`

**return** $\mathcal{P}, \mathcal{F}, \mathcal{U}$

---

levelset estimation [17, 18, 19] has also been applied to increase sample efficiency in discretized coverage-based testing [8]. On the other hand, [9] leverages adaptive discretization resolutions.

### 2.2 Adversarial Example Generation

Adversarial example generation is a widely studied approach to generate difficult exaples [20, 21, 22, 23, 24, 25], which have been applied to image classification [26], natural language processing [27], and self-driving [10]. In parameterized scenario testing, this is done by optimizing parameters via black-box optimization algorithms such as Bayesian Optimization [10, 28, 29, 30, 31] or Reinforcement Learning [32, 33] to generate challenging safety-critical scenarios. These methods optimize for the most adversarial and do not cover the parameter space. On the other hand, [34] showed that BO with the GP-LCB acquisition function can validate that an AV passes all scenario variations if the LCB is positive everywhere and a failure isn't discovered. This achieves coverage but only in the case where the AV is perfect. Acquisition functions for adversarial example generation can also be used to sample tests in our framework. However, this oversamples severe failures, whereas oversampling on the pass/fail boundary is more efficient for coverage-based testing.

## 3 Scalable Coverage-Based Testing

In this section, we introduce our formulation which leverages probabilistic models to partition the parameter space into pass, fail, and unknown regions. Following our formulation, we introduce our testing framework GUARD that leverages Gaussian Processes (GPs), levelset estimation, and learnable GP kernels. An overview can be found in Figure 1 and the algorithm in Algorithm 1.

### 3.1 Problem Formulation

The AV software stack is often tested on parameterized scenarios commonly known as *logical scenarios* [13, 6], which are the standard in the self-driving industry. Each logical scenario features a set of $d$ configurable parameters $\boldsymbol{\theta} \in \mathbb{R}^d$, where each specific configuration of the parameters results in a concrete test. The scenario parameters are bounded [13, 15], i.e., $\theta_i \in [a_i, b_i]$, making the entire parameter search space a closed set $\boldsymbol{\Theta}$ in $\mathbb{R}^d$. We assume that a simulation test outputs a scalar measure of safety (e.g. minimum distance to another agent). Mathematically, let $f^* : \boldsymbol{\theta} \times \mathcal{A} \to \mathbb{R}$ be the test function which takes as input test parameters $\boldsymbol{\theta}$, autonomy system $\mathcal{A}$ and outputs a real-valued scalar $f^*(\boldsymbol{\theta}; \mathcal{A})$. We omit $\mathcal{A}$ to use $f^*(\boldsymbol{\theta})$ for brevity. In compliance with regulatory demand [7], a binary pass or fail $y = \mathbf{1}[f^*(\boldsymbol{\theta}) \geq \gamma]$ is computed using a threshold $\gamma$. It follows that we can denote regions in the parameter space $\boldsymbol{\Theta}$ where the system passes and fails as

$$\mathcal{P}^* = \{\boldsymbol{\theta} \in \Theta, f^*(\boldsymbol{\theta}) \geq \gamma\}. \tag{1}$$

$$\mathcal{F}^* = \{\boldsymbol{\theta} \in \Theta, f^*(\boldsymbol{\theta}) < \gamma\}. \tag{2}$$

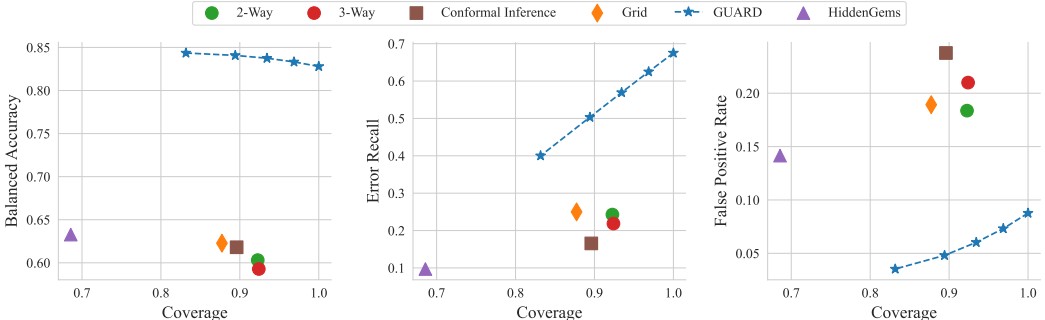

Figure 2: GUARD significantly outperforms other testing frameworks. GUARD also a sweepable confidence threshold to a trade-off between coverage and the other metrics.

Then, the testing process can be thought of as sequencing a finite set of tests $\{\boldsymbol{\theta}_1, \ldots, \boldsymbol{\theta}_N\}$, observing their outcomes $\{f^*(\boldsymbol{\theta}_1), \ldots, f^*(\boldsymbol{\theta}_N)\}$ and estimating $\mathcal{P} \approx \mathcal{P}^*$, $\mathcal{F} \approx \mathcal{F}^*$. Estimating pass/fail across a continuous parameter domain using a finite set of test points inevitably leads to estimation errors, especially if the number of concrete tests are limited (e.g. due to testing resource constraints). At the same time, estimation errors and false positive passes in particular can be detrimental to safety.

Thus, we design an uncertainty-aware formulation where the framework quantifies the confidence of its estimation. This allows the framework to say it is uncertain instead of predicting a pass/fail outcome with insufficient information. Specifically, we compute the probability that the system will pass or fail at any point in the parameter space. To achieve this, we model $f^*(\boldsymbol{\theta})$ with a random variable $f(\boldsymbol{\theta})$. Under this random variable, the estimated pass and fail regions can be defined as

$$
\begin{aligned}
\mathcal{P} &= \{\boldsymbol{\theta} \in \boldsymbol{\Theta}, P(f(\boldsymbol{\theta}) \geq \gamma) \geq \alpha\}. \\
\mathcal{F} &= \{\boldsymbol{\theta} \in \boldsymbol{\Theta}, P(f(\boldsymbol{\theta}) < \gamma) \geq \alpha\}.
\end{aligned}
\tag{3}
$$

with respect to confidence threshold $\alpha$. It follows that the unknown region can be obtained as $\mathcal{U} = \boldsymbol{\Theta} - \mathcal{P} - \mathcal{F}$, where more information is required to make a prediction. Note that the size of $\mathcal{P}$ and $\mathcal{F}$ increase monotonically as $\alpha$ decreases. The configurable confidence threshold $\alpha$ allows the testing framework to control the tradeoff between the quality and quantity of predictions.

### 3.2 Gaussian Process

As outlined in Equation 3, our formulation requires a probabilistic estimate $f(\boldsymbol{\theta})$ for the ground truth test function $f^*(\boldsymbol{\theta})$. We adopt GPs as they perform well in low-data regimes, making them suitable in our setting as the number of concrete tests are limited. Additionally, GPs are non-parametric and make less assumptions about the function being modeled, which is important in allowing our testing framework to scale to a wide variety of logical scenarios.

A GP is a collection of random variables in which every subset is assumed to distributed from a multivariate Gaussian. Let $X = [\boldsymbol{\theta}_1 \ldots \boldsymbol{\theta}_N] \in \mathbb{R}^{n \times d} \sim \mathcal{N}(\mu, \Sigma)$ be random variables from N test samples and $Y = [f^*(\boldsymbol{\theta}_1) \ldots f^*(\boldsymbol{\theta}_N)]$ be the corresponding scalar outputs of $f^*$. To estimate test outcomes, we model the distribution over the real-valued output of the metric $P(f(\boldsymbol{\theta}))$. Specifically, the value of an unseen datapoint is estimated by the conditional posterior distribution

$$
P(f(\boldsymbol{\theta})|\boldsymbol{\theta}, X, Y) \sim \mathcal{N}(\mu(\boldsymbol{\theta}), \sigma^2(\boldsymbol{\theta}))
\tag{4}
$$

$$
\mu(\boldsymbol{\theta}) = k(\boldsymbol{\theta}, X)^T (K + \epsilon^2 I)^{-1} Y
\tag{5}
$$

$$
\sigma^2(\boldsymbol{\theta}) = k(\boldsymbol{\theta}, \boldsymbol{\theta}) - k(\boldsymbol{\theta}, X)^T (K + \epsilon^2 I)^{-1} k(\boldsymbol{\theta}, X).
\tag{6}
$$

Here $k(\cdot, \cdot)$ is the kernel function which provides a similarity measure between two GP variables, $k(\boldsymbol{\theta}, X) = [k(\boldsymbol{\theta}, \boldsymbol{\theta}_i), \ldots, k(\boldsymbol{\theta}, \boldsymbol{\theta}_N)]$, $K_{ij} = k(\boldsymbol{\theta}_i, \boldsymbol{\theta}_j)$, and $\epsilon$ models noise in the observed values $Y$. Under the posterior distribution of $f(\boldsymbol{\theta})$, the probability of observing a value above or below a certain threshold can be computed. Specifically, the probability of observing a value greater than the pass-fail threshold $\gamma$ is

$$
P(f(\boldsymbol{\theta}) \geq \gamma) = \Phi\left(\frac{\mu(\boldsymbol{\theta}) - \gamma}{\sigma(\boldsymbol{\theta})}\right).
\tag{7}
$$

where $\Phi$ is the CDF of the normal distribution. The probability of observing a value under $\gamma$ can be computed in a similar fashion. An illustration of this can be seen in Figure 1.

| Method | Coverage(%) | Bal. Acc(%) | Pos Acc(%) | Neg Acc(%) | Err. Recall(%) | FPR(%) |
|---|---|---|---|---|---|---|
| Grid | $87.7 \pm 0.4$ | $62.3 \pm 0.1$ | $97.4 \pm 0.03$ | $27.2 \pm 0.2$ | $25.0 \pm 0.3$ | $18.9 \pm 0.2$ |
| 2-way [16] | $92.2 \pm 0.2$ | $60.3 \pm 0.6$ | $96.0 \pm 0.1$ | $24.6 \pm 1.2$ | $24.3 \pm 1.2$ | $18.2 \pm 0.4$ |
| 3-way [16] | $92.4 \pm 1.0$ | $59.7 \pm 1.7$ | $96.2 \pm 0.3$ | $23.2 \pm 3.7$ | $22.5 \pm 3.6$ | $20.9 \pm 1.3$ |
| HiddenGems [8] | $72.8 \pm 0.6$ | $66.2 \pm 2.2$ | $98.3 \pm 0.1$ | $33.7 \pm 4.4$ | $14.8 \pm 2.4$ | $16.0 \pm 0.4$ |
| Conformal Inference [9] | $89.6 \pm 0.4$ | $61.8 \pm 0.8$ | $97.0 \pm 0.7$ | $26.7 \pm 1.4$ | $16.6 \pm 1.5$ | $23.8 \pm 0.9$ |
| GUARD | $\mathbf{94.3 \pm 0.9}$ | $\mathbf{84.2 \pm 5.3}$ | $\mathbf{99.2 \pm 0.2}$ | $\mathbf{70.3 \pm 10.6}$ | $\mathbf{58.9 \pm 5.3}$ | $\mathbf{5.89 \pm 3.1}$ |

Table 1: Benchmarking GUARD against baselines

## 3.3 Test Generation

Note that precisely modelling $f^*$ when the output is far from $\gamma$ adds little value since it won't affect the pass or fail outcome. In contrast, it's important to accurately model regions near the $\gamma$-levelset. Hence, we leverage levelset algorithms to efficiently upsample points near the boundary. Specifically, we adopt *Straddle* [35] as it directly integrates GPs, and alternative GP-levelset algorithms operate on finite input domains [17, 8, 18]. *Straddle* iteratively queries test points based on an exploration incentive promoting points with high variance and an exploitation incentive promoting points close to the $\gamma$-levelset. Specifically, these two incentives are captured by the acquisition function

$$h_t(\boldsymbol{\theta}) = \beta \sigma_t(\boldsymbol{\theta}) - |\mu_t(\boldsymbol{\theta}) - \gamma| \tag{8}$$

where $\beta$ is a weighting coefficient balancing the first exploration term and the second exploitation term. At each iteration, we solve a lightweight maximization problem to find the next query point $\boldsymbol{\theta}_t = \operatorname{argmax}_{\boldsymbol{\theta}} h_t(\boldsymbol{\theta})$. We start by sampling $P$ initial random points from $\boldsymbol{\Theta}$. Then since $h_t(\boldsymbol{\theta})$ is differentiable with respect to $\boldsymbol{\theta}$, the top $Q$ candidates are improved via gradient-based optimization. This optimization is inexpensive since the cost of running the GP model and performing backpropagation to evaluate $\mu(\boldsymbol{\theta}), \sigma(\boldsymbol{\theta})$ is negligible compared to running simulation to evaluate $f^*(\boldsymbol{\theta})$. Finally, the concrete test corresponding to $\boldsymbol{\theta}_t$ is executed in simulation to obtain $f^*(\boldsymbol{\theta}_t)$. The observation $(\boldsymbol{\theta}_t, f^*(\boldsymbol{\theta}_t))$ is then added to the GP model to update the GP posterior.

## 3.4 Automatic Kernel Tuning

In addition to observed test points, the posterior distribution outlined in Equation 5 and 6 also depend heavily on the kernel function $k(\cdot, \cdot)$. Thus, the estimates $\mathcal{P}$ and $\mathcal{F}$ are sensitive to the kernel function parameters which we denote as $\boldsymbol{\kappa}$. At the same time, tuning kernel parameters can be tedious and time consuming [8]. Moreover, in our task, each individual scenario has different parameters with varying scales and effects on the final output, requiring different kernels to accurately model different scenarios. This makes manually tuning kernels infeasible for large scale testing.

To circumvent this issue, we first normalize the scenario parameter space $\boldsymbol{\Theta}$ to the unit hypercube $[0, 1]^d$ to address parameters with different scales. Furthermore, instead of manually tuned and fixed kernel parameters, we optimize the kernel parameters to maximize the marginal likelihood of the observations. In particular, at iteration $t$, we update the kernel parameters $\boldsymbol{\kappa}$ towards

$$\boldsymbol{\kappa}_t = \operatorname*{argmax}_{\boldsymbol{\kappa}} P(f(\boldsymbol{\theta}_1) \ldots f(\boldsymbol{\theta}_t) \mid \boldsymbol{\theta}_1 \ldots \boldsymbol{\theta}_t; \boldsymbol{\kappa}). \tag{9}$$

The marginal likelihood is differentiable with respect to the kernel parameters $\boldsymbol{\kappa}_t$ and we perform gradient-based optimziation every $K$ iterations. The kernel learning process can be prone to overfitting to a small initial batch of observations. Therefore, we perform an initial sampling step where we query $M$ tests using only the variance term $\sigma_t(\boldsymbol{\theta})$ of the acquisition function outlined in Equation 8.

# 4 Experiments

We first introduce the test scenarios, evaluation metrics, and relevant baselines. Then, we demonstrate the effectivness of GUARD and ablate our design choices. Finally, we show how GUARD can be used in practice to benchmark autonomy systems and catch regressions. Additional experiments are included in Appendix A.

## 4.1 Experimental Setup

**Simulation Test Scenarios:** We use a suite of 10 logical scenarios designed by expert testing engineers and based on guidelines [36] prescribed by the National Highway Traffic Safety Association (NHTSA). The scenarios are executed in a high-fidelity closed-loop simulator and use a combination of scripted and reactive actors. The scripted actors execute parameterized maneuvers to stress

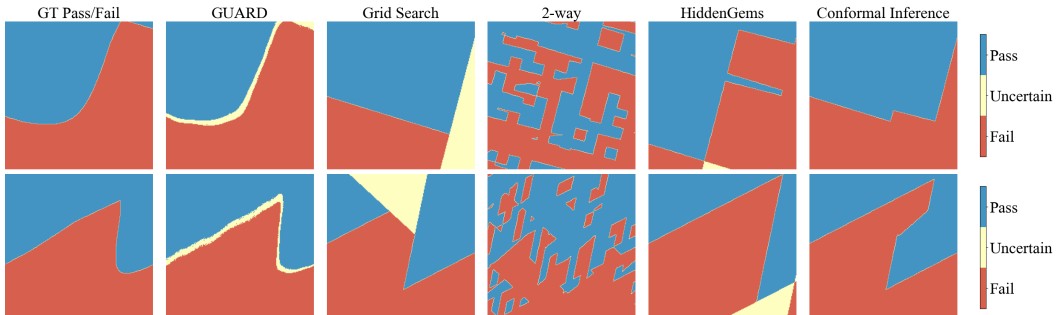

Figure 3: 2D slice of 5D parameter space. GUARD performs better by avoiding discretization.

test the AV - e.g. cut-ins. The reactive actors act as realistic background traffic and are controlled using the Intelligent Driver Model [37]. Each scenario has $d = 5$ parameters and descriptions of the scenarios are provided in Appendix B.

We evaluate the PLT motion planner [38] using minimum distance(m) to collision as the safety metric and choosing the safety threshold as $\gamma = 1.0$. We use $N = 1000$ test samples for experiments unless otherwise specified. In the following experiments, we repeat 5 trials for each scenario to collect mean and standard deviation of metrics. We report the average mean and standard deviation across the 10 scenarios. To support large scale experimentation, we construct an offline dataset by evaluating each scenario space over a fine grid to create an approximation via linear interpolation.

**Metrics:** To evaluate performance, we propose several metrics which capture important aspects of testing for safety and autonomy development.

- *Coverage* is the percentage of the search space $\Theta$ that is covered by the estimates $\mathcal{P}$ and $\mathcal{F}$:

$$\text{coverage} = \frac{|\mathcal{P} \cup \mathcal{F}|}{|\Theta|}. \tag{10}$$

- *Balanced Accuracy* evaluates how accurate the estimates $\mathcal{P}$ and $\mathcal{F}$ are and also addresses class imbalance due to the fact that the scenario space is dominated by passes:

$$\text{balanced acc} = \underbrace{\frac{|\mathcal{P} \cap \mathcal{P}^*|}{|\mathcal{P}^*| \cap (|\mathcal{P} \cup \mathcal{F}|)}}_{\text{positive acc}} + \underbrace{\frac{|\mathcal{F} \cap \mathcal{F}^*|}{|\mathcal{F}^*| \cap (|\mathcal{P} \cup \mathcal{F}|)}}_{\text{negative acc}} \tag{11}$$

- *Error Recall* measures the percentage of the failures the testing framework can recall, which is very useful for autonomy development. This computed as:

$$\text{error recall} = \frac{|\mathcal{F} \cap \mathcal{F}^*|}{|\mathcal{F}^*|}. \tag{12}$$

- *False Positive Rate* is the percentage of predicted passes that are incorrect, which is crucial to safety as incorrectly predicting the system to be safe can lead to accidents. This is computed as:

$$\text{false positive rate} = \frac{|\mathcal{P} \cap \mathcal{F}^*|}{|\mathcal{P}|}. \tag{13}$$

Note that some of these quantities such as $|\mathcal{P}^*|$ cannot be computed exactly. Therefore, we randomly sample $20,000$ points in the scenario parameter space to estimate the terms in the above equations.

**Baselines:** We compare GUARD with several baselines outlined in Section 2. These include traditional testing methodologies [16, 14, 15] and works proposed in the literature [8, 9].

- *Grid Search* [15, 14] discretizes continuous parameters, and uses one discrete test point to represent each bin. Given budget of $N$ test samples which cannot cover all bins, we sample a subset of the bins at random. In our scenarios with $d = 5$ dimensions, we use discretization resolution of 4.
- *T-way testing* [16] leverages the assumption that most faults are caused by interactions of at most $t$ parameters. This method draws test samples such that for every $t$-parameter subset, all possible combinations of parameter values can be found in at least one test. In our experiments, we consider $t = 2, 3$ and choose a discretization resolution of 30 for $t = 2$ and 12 for $t = 3$.

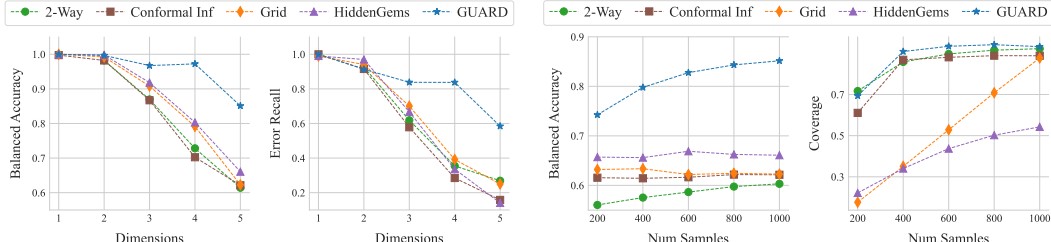

Figure 4: Scaling with scenario dimensionality.

Figure 5: Scaling with testing budget.

- *HiddenGems* [8] employs levelset estimation algorithms to more efficiently infer pass/fail outcomes of the bins. The authors use a 33x33 discretization for a 2-D parameter space. However, this resolution becomes infeasibly expensive with $d = 5$ parameters. We adopt a resolution of 6 and further analyze the exponentially increasing runtime in Appendix A.
- *Conformal Inference* [9] uses conformal inference on samples in each bin to determine the outcome. Bin sizes are not fixed, since this method chooses how the parameter space is partitioned.

**Implementation Details:** The initial exploration batch has size $M = 200$. Our GP kernel is a product composition of separate Matern32 kernels [39] for each parameter dimension. The kernel lengthscale is initialized to $0.02$ and variance initialized to the variance of the initial exploration batch. The kernel is fitted every 50 samples using Adam [40] with learning rate $0.01$ for 100 steps. Our acquisition function uses $\beta = 0.5$. During acquisition maximization, we sample $P = 600$ candidates improve the top $Q = 30$ candidates using Adam with learning rate $0.01$ for 30 steps.

## 4.2 Experiment Results

**Testing Performance:** We demonstrate the effectiveness of GUARD in Figure 2 where we plot test coverage versus balanced accuracy, error recall, and false positive rate (FPR). First, note that we can adjust the trade-off between coverage and the other metrics in GUARD by adjusting the confidence threshold $\alpha$ as outlined in Equation 3. This is useful in controlling how conservative we are in testing. Along the trade-off curve, we can observe how reliable the predictions are at different probability thresholds. Compared to other methods, GUARD performs better across all metrics. We provide numerical results in Table 1 where we set $\alpha = 0.7$ for direct comparison.

To illustrate how GUARD achieves better metrics, we visualize the outputs in Figure 3, using the technique proposed in [41] to visualize a 2D slice of the 5D parameter search space. Specifically, we sample one boundary point and multiple slices on the boundary point, choosing the slice with the highest variance. On each slice we visualize the ground truth and predicted pass/fail regions. Compared to other methods, GUARD is more accurate as it's not limited by discretization.

**Scalability:** We evaluate GUARD against baselines on lower dimensional functions to analyze scalability with the number of scenario parameters. The 10 test scenarios have $5 - c$ parameters fixed to reduce the dimension to $c$. We increase the discretization resolution of baselines at lower resolutions, choosing the resolutions to maintain a similar amount of bins as in the 5-D scenarios. Figure 4 shows that baselines perform well for 1-D and 2-D functions where they can afford a high resolution. As dimensionality increases the baselines degrade significantly compared to GUARD.

**Sample Efficiency:** In Figure 5, we evaluate the models using a varying number of test samples $N = [200, 400, 600, 800, 1000]$ and show that GUARD outperforms the alternative methods with fewer test samples as well. With increasing testing budget, GUARD gradually increases in accuracy and coverage. Other approaches which rely on discretization grow in coverage. However, their accuracy is heavily limited by the discretization resolution and improves slowly or not at all.

**Ablations:** We ablate some design choices of GUARD in Table 2. First, without the initial exploration phase, there is a noticeable drop across the metrics. Next, we consider removing online kernel learning, parameter normalization, and both. Removing either leads to drops in performance while removing both leads to degenerate solutions which predict pass everywhere. This is expected as without normalization or learnable lengthscales, a fixed lengthscale cannot model parameters with different scales (i.e. velocity $\in [20, 30]m/s$, road curvature $\in [-0.002, 0.002]$). Finally, we ablate the choice of the GP kernel. As described in Section 4.1, GUARD uses a product of Matern32 kernels on each parameter because each parameter affects the function output differently. Using a single kernel which doesn't consider each parameter individually degrades performance as expected.

| Exp | Norm | Learn | Prod | Coverage(%) | Bal. Acc(%) | Pos. Acc(%) | Neg. Acc(%) | Err. Recall(%) | FPR (%) |
|---|---|---|---|---|---|---|---|---|---|
| ✓ | ✓ | ✓ | ✓ | **94.3 ± 0.9** | **84.2 ± 5.3** | 99.2 ± 0.2 | **70.3 ± 10.6** | **58.9 ± 5.3** | 5.9 ± 3.1 |
| — | ✓ | ✓ | ✓ | 92.2 ± 1.9 | 82.4 ± 4.9 | **99.4 ± 0.2** | 65.4 ± 10.0 | 52.1 ± 8.8 | 7.5 ± 3.3 |
| ✓ | — | ✓ | ✓ | 93.2 ± 1.2 | 83.7 ± 2.3 | 99.0 ± 0.2 | 68.3 ± 4.6 | 56.3 ± 4.4 | **5.8 ± 1.6** |
| ✓ | ✓ | — | ✓ | 92.3 ± 0.8 | 84.1 ± 1.9 | 98.7 ± 0.2 | 69.7 ± 3.8 | 48.6 ± 3.9 | 6.2 ± 1.0 |
| ✓ | — | — | ✓ | 0.04 ± 0.03 | – | – | – | – | – |
| ✓ | ✓ | ✓ | — | 93.3 ± 1.9 | 81.4 ± 4.1 | 99.0 ± 0.2 | 63.8 ± 8.3 | 51.6 ± 8.1 | 7.3 ± 1.9 |

Table 2: Ablating design choices of GUARD

## 4.3 Using GUARD In Practice

Different AV systems may have different failure modes and find different scenarios challenging. Since GUARD is agnostic to the system under test, it can evaluate any AV system to discover where it passes and fails across. This is useful during development to identify regressions between two iterations of the AV. To demonstrate this, we use PLT as a reference planner and introduce a variation adjusted to make the planner more *aggressive*.

First, we evaluate both planners on our test scenarios to measure the aggregate pass and fail rates. As shown in Table 4.3, GUARD correctly identifies the aggressive planner as less safe. Furthermore, we can triage the regressions by visualizing the pass/fail landscape which we show in Figure 6 below.

| Planner | Pass Rate | Fail Rate |
|---|---|---|
| Ref | 69.8% | 18.5% |
| Agg | 66.9% | 24.0% |

Table 3: Comparing planners.

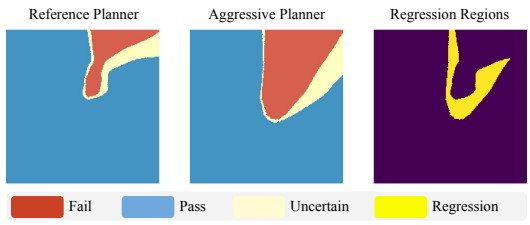
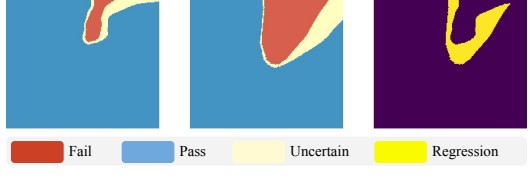

Figure 6: Identifying regression regions - reference planner passes, aggressive planner fails.

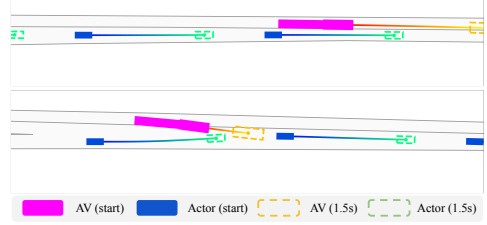

Figure 7: Regression example. **Top**: Reference. **Bottom**: aggressive planner collides.

To aid autonomy development, we can sample any point from the regression region to yield a concrete test. We show one example of such a test in Figure 7. In this test, the ego truck attempts to change into a neighboring lane while actors from an on-ramp are merging. The aggressive planner does not lane change safely which ultimately leads to a collision. Revealing all of these regressions in an automatic and scalable way is invaluable to autonomy development.

## 5 Limitations

The scope of testing is limited since we assume the pass/fail result is thresholded on a single measure of safety, whereas it can be a combination of multiple measures. We also did not consider discrete scenario parameters which can be categorical (e.g. vehicle type) or numerical (e.g. number of lanes). Future works can incorporate these considerations to build a more complete testing framework. In addition, GPs relies on the assumption that the landscape of the safety measure is smooth enough to be modelled by the GP kernel. Investigating alternative probabilistic models such as neural GPs [42] and probabilistic SVMs [43] is another exciting direction. Finally, the impact of our work and simulation testing in general is highly dependent on the fidelity of the simulator. Reducing the sim2real gap remains an open problem [44, 45, 46, 47].

## 6 Conclusion

This paper tackles coverage-based testing of AVs in parameterized scenarios. We formulate the problem as sequencing a finite set of tests to estimate the probability of passing or failing across the entire parameter space. Based on this formulation, we propose a testing framework GUARD that efficiently samples tests to cover the parameter space and accurately evaluate the AV's performance. This framework can be used in practice with functional safety experts defining a comprehensive set of safety requirements and a parameterized operational design domain (ODD). GUARD can automatically generate concrete tests and validate the AV meets these requirements across the ODD. Our work contributes to streamlined autonomy development, safety validation, and is ultimately a step towards safely deploying autonomous vehicles.

**Acknowledgements**

The authors would like to thank the anonymous reviewers for their valuable feedback and suggestions to improve the paper. We would also like to thank Paul Spriesterbach and Andre Strobel for their input on how our work relates to functional safety guidelines. Finally we would also like to thank Wenyuan Zeng and Sean Segal for their contributions in brainstorming and suggestions on improving the manuscript.

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

# A   Additional Experiments

**Ablation on GP Kernels:**    While GUARD requires minimal tuning of GP kernel parameters due to online kernel learning, the *type* of kernel is still a design choice. In GUARD we employ Matern32 kernels to model each parameter and take the product of individual kernels to obtain the final kernel. In Table 4 we conduct an ablation that also considers 1) RBF kernels, 2) Using sum instead of product, and 3) no composition, using one RBF or Matern32 kernel for all parameters. We first find that without modeling each parameter separately, the results are poor. Also, using a sum operation to aggregate kernels is ineffective as well. The best results are obtained from a product of RBF or Matern32 kernels. We choose product of Matern32 as the lower false positive rate is critical for ensuring safety. Furthermore, this choice also exhibits much lower variance in the metrics, making it more reliable for testing.

**BO Acquisition Functions:**    As mentioned in Section 2 of the main manuscript, many adversarial example generation methods use Bayesian Optimization(BO) and employ acquisition functions that minimize some adversarial objective. Since BO also leverages GPs and active learning similar to GUARD, we can adopt these acquisition functions. In particular we use lower confidence bound (LCB), expected improvement (EI), and probability of improvement (PI). For EI and PI we adjust them to decrease the objective (expected decrease and probability of decrease). The results in Table 5 show that these alternatives can achieve strong results but is not as effective as using the Straddle acquisition function. This is because cost minimization oversample very negative points and Straddle oversamples boundary points, but the former is more useful in partioning pass / fail regions. The performance is however very close, and we hypothesize this is because the parameter space is dominated by positive (passes). With very small negative regions, severe negatives are close to boundary points.

**Runtime Comparison**    A major implementation detail regarding our baselines is the discretization resolution. For grid search and $t$-way testing, there is a direct tradeoff between coverage and balanced accuracy, error recall, false positive rate. Thus, we select a resolution which can cover most of the parameter space. For HiddenGems, the resolution does not affect coverage since it uses a levelset estimation algorithm to determine which bins are covered. The limitation for our choice of resolution of 6 is due to how the computation budget increases exponentially with resolution. This is because the method requires running the GP on all discretized bins at every iteration. And while running the GP a single time is neglible compared to running simulation, the exponential scaling makes the GP queries a bottleneck at higher resolutions. As we show in Figure A, at this resolution the runtime roughly equals GUARD.

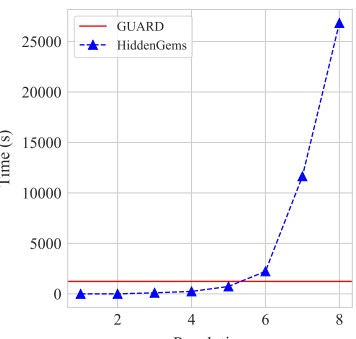

Figure 8: Runtime

| Kernel Type | Composition | Coverage(%) | Bal. Acc(%) | Pos Acc(%) | Neg Acc(%) | Err. Recall(%) | FPR(%) |
|---|---|---|---|---|---|---|---|
| RBF | - | $96.1 \pm 0.8$ | $80.2 \pm 5.7$ | $98.1 \pm 0.5$ | $62.4 \pm 11.2$ | $56.0 \pm 10.6$ | $10.6 \pm 3.3$ |
| Matern32 | - | $87.3 \pm 1.5$ | $78.5 \pm 4.2$ | $\mathbf{99.3 \pm 0.2}$ | $57.7 \pm 8.4$ | $36.8 \pm 7.7$ | $6.15 \pm 1.1$ |
| RBF | Sum | $88.0 \pm 6.7$ | $58.6 \pm 3.2$ | $89.1 \pm 11.4$ | $28.2 \pm 11.5$ | $21.4 \pm 12.0$ | $20.2 \pm 5.7$ |
| Matern32 | Sum | $90.4 \pm 3.3$ | $58.4 \pm 3.3$ | $88.0 \pm 10.7$ | $28.7 \pm 13.7$ | $23.6 \pm 12.7$ | $21.0 \pm 3.1$ |
| RBF | Product | $\mathbf{96.8 \pm 0.7}$ | $83.3 \pm 7.0$ | $97.9 \pm 0.6$ | $68.7 \pm 13.9$ | $\mathbf{62.6 \pm 13.1}$ | $7.83 \pm 4.0$ |
| Matern32 | Product | $94.3 \pm 0.9$ | $\mathbf{84.2 \pm 5.3}$ | $99.2 \pm 0.2$ | $\mathbf{70.3 \pm 10.6}$ | $58.9 \pm 5.3$ | $\mathbf{5.89 \pm 3.1}$ |

Table 4: Ablation of different kernel choices for GUARD

| Aquisition Fn. | Coverage(%) | Bal. Acc(%) | Pos Acc(%) | Neg Acc(%) | Err. Recall(%) | FPR(%) |
|---|---|---|---|---|---|---|
| PI | $90.7 \pm 1.6$ | $82.2 \pm 3.9$ | $99.4 \pm 0.1$ | $64.9 \pm 7.8$ | $49.0 \pm 7.5$ | $8.23 \pm 2.4$ |
| EI | $91.8 \pm 1.3$ | $82.6 \pm 2.3$ | $99.4 \pm 0.2$ | $65.7 \pm 0.5$ | $50.7 \pm 5.2$ | $6.99 \pm 1.4$ |
| LCB | $91.5 \pm 1.5$ | $83.4 \pm 1.8$ | $\mathbf{99.6 \pm 0.1}$ | $67.3 \pm 3.6$ | $52.4 \pm 4.7$ | $5.99 \pm 1.4$ |
| Straddle | $\mathbf{94.3 \pm 0.9}$ | $\mathbf{84.2 \pm 5.3}$ | $99.2 \pm 0.2$ | $\mathbf{70.3 \pm 10.6}$ | $\mathbf{58.9 \pm 5.3}$ | $\mathbf{5.89 \pm 3.1}$ |

Table 5: Comparison of cost minimization acquisition functions versus Straddle acquistion function.

## B  Test Scenarios

We describe the 10 logical scenarios used in our experiments here:

1. **Actor cut-in**: An actor starts in an adjacent lane to the SDV and performs a cut-in maneuver.
2. **Shoulder actor cut-in**: The SDV drives next to the road shoulder. An actor on the shoulder cuts in front of the SDV.
3. **Actor overtake cut-in**: An actor starts behind the SDV, then rapidly accelerates to overtake the SDV and cut-in front of it.
4. **Actors merging:** The SDV is driving on a merge lane and there are multiple actors merging in.
5. **Lead actor braking**: An actor starts in front of the SDV and then brakes.
6. **Lane change**: The SDV attempts to lane change when there is an actor in ithe target lane.
7. **Lane change beside merge**: The SDV attempts to lane change into a merge lane and actors are also merging into that lane.
8. **Lane merge**: The SDV is merging from an on-ramp, and there is an actor in the merge lane that is very close to the SDV.
9. **Lane merge multiple actors**: The SDV is merging from an on-ramp and there are multiple actors in the merge lane,.
10. **Lane merge parallel on-ramp**: The SDV is merging from a parallel on-ramp and there is an actor in the merge lane that is very close to the SDV.

