# OpenReview forum: "Towards Scalable Coverage-Based Testing of Autonomous Vehicles"
_robot-learning.org/CoRL/2023/Conference — CoRL 2023 Poster_

### Official Review · Reviewer_QJej · 2023-07-15

**Confidence:** 3
**Originality:** Good
**Technical Quality:** Good
**Clarity Of Presentation:** Good
**Impact:** 2

**Recommendation:**

Weak Accept: I recommend accepting the paper, but will not argue for my recommendation if the majority of other reviewers have a different opinion.

**Review:**

Strengths:
* The paper provides many ablations of the proposed method, and comparisons quantitative and qualitative with other methods.
* The paper’s approach – using Gaussian Processes to infer the likelihood of failure under some scenario parameters – is, to the best of my knowledge, an original approach to finding failure / success scenarios.

Weaknesses:
* The mathematical clarity could be improved. Specifically, in the Gaussian Process section it was difficult to parse what was being modeled / inferred and from what data was being used to train the GP.  See more detailed comments below.
* It was challenging to understand one of the key contributions – computational efficiency – of the proposed method from the main text. I would recommend including a metric like compute time on a standard computer for all methods. Additionally, even though efficiency is one of the proposed advantages of the method, the parameter dimensional still appears quite low – 5D at max – in the experiments.
* I would like to see how the proposed method compares in terms of scalability and search performance to adaptive stress testing methods that utilize Deep RL / Monte Carlo Tree Search. These approaches seem to have a much stronger potential to scale to large state / parameter spaces than GPs. Detailed comments and specific methods are mentioned in the rebuttal section.

Overall comments:
* “The observed test samples $X = [\theta_1;... ; \theta_N], Y = [f^*(\theta_1)...; f^*(\theta_N)]$ then become observations of GP random variables” is a difficult sentence to parse, as is the math in section 3.2. Is X the observation or is Y the observation? And what is the final distribution you are modeling with the GP? It would be helpful to clearly write out 1) what are the random variables and how are they distributed (e.g., let $X = [\theta_1, …\theta_N] \in R^{n \times d} \sim N(\mu_\theta, \Sigma_\theta)$ be a random variable of N parameter values. Let $Y = (f^*(\theta_1), …f^*(\theta_N)) \in R^n$ be the corresponding scalar outputs of $f^*$ of each $\theta$, …), and 2) what distribution is being modeled (e.g., saying something like `we seek the conditional distribution $P(Y | X)$ which models the probability of test outputs given a set of parameter values. We infer this distribution from…and model it as a Gaussian process with mean $\mu$, and covariance $\Sigma$…’’).
* It is also unclear why the sets $P*, F*, P$ and $F$ in equation 3 are defined at the very start since most of the focus of this paper is on maintaining the full distribution, and these sets are only used in the evaluation metrics and not during the core of the method. It would be clearer to move these to Section 4.
* In the main text, Section 4, it was difficult to understand 1) what simulator was being used?, 2) how was the data simulated for all the road agents? In particular, 2) is of concern because if the data for non-ego road agents was drawn from a static dataset, then the results would look different than from a road agent simulator where the agents can adapt/react to the ego vehicle.
* The text writes  “Executing a test in simulation outputs continuous measures of safety or other performance metrics… $f^*(\theta, A) \in R$”. However, as noted in the Limitations section, in most AV driving scenarios, there is more than one performance metric – for example, one may be interested in a safety metric (e.g. the minimum distance between the vehicles) and a progress metric (e.g., time-to-destination). It would be helpful to make this assumption more clear upfront. For example, “We assume that a simulation test outputs a scalar indicating the level of safety (e.g., minimum distance to another roach agent). Mathematically, let  $f^*: \theta \times A \rightarrow R$ be the test function which takes as input the simulation parameters, and autonomy software A and outputs a real-valued scalar.”

**Quality Of The Limitations Section:**

Limitations are addressed clearly

**Questions For Rebuttal:**

* In section 3.2, the writing and notation made it hard to tell if the goal was to infer a distribution over test parameters \theta (i.e., $P(\theta)$), a distribution of pass-or-fail (i.e. $P(y)$), or a distribution over the real-valued output of the metric (i.e. $P( f(\theta) )$). Which distributions are being learned by the GP?
* How does this work compare (in terms of scalability, ability to find failure scenarios, etc.) to adaptive stress testing approaches like [1] and [2] which utilize RL and monte carlo tree search?
[ 1 ] Lee, Ritchie, et al. "Adaptive stress testing: Finding likely failure events with reinforcement learning." Journal of Artificial Intelligence Research 69 (2020): 1165-1201.
[ 2 ] Koren, Mark, et al. "Adaptive stress testing for autonomous vehicles." 2018 IEEE Intelligent Vehicles Symposium (IV). IEEE, 2018.
* What do the parameters mean in this experimental environment? How was the space of $\Theta$ constructed? Also, why was $d = 5$ parameters chosen in these experiments?
* The efficiency results only report how accuracy and error are impacted by dimensionality of the parameter space. However,I would like to see compute time comparisons of each method for the same scenario to understand the compute efficiency.

**Robotics Focus:**

Relevant but unlikely to deploy to hardware in near future

**Summary Of Paper:**

This paper focuses on testing autonomous vehicle software in simulation. The key idea is to infer the probability of a test output (e.g., minimum distance between road agents in the scenario) given some scenario parameters in a scalable but high-quality way. The paper uses a Gaussian Process to infer this probability and uses a previously developed sampling scheme to prioritize data points at the boundary of a pass / fail of the test (where the pass / fail boundary is determined a priori by, for example, a regulatory agency). The paper compares the proposed approach to an exhaustive discretization approach and three other baseline for a 5D scenario parameter space. It shows favorable performance in terms of accuracy, error, and coverage metrics.

**Summary Of Recommendation:**

While I think this paper focuses on a valuable problem and has an interesting approach, I have two concerns which lead me to recommend a weak reject.  First, the paper's approach of using Gaussian Processes is not benchmarked against SOTA learning-based approaches. This is specifically relevant for two reasons: 1) it is well-known that GPs scale poorly with the number of observations, and one of the major claims of this paper is a more scalable framework, and 2) other SOTA methods mentioned in my review that, for example, use Deep RL, have better potential for scalability and are highly relevant to this conference. Secondly, the clarity of the writing and experimental section should be improved to shed light on this paper's approach and relevance for autonomous driving (e.g, simulator used, models for the road agents during testing, what the scenario parameters model, how are these scenario params chosen, etc.).

---

### Official Review · Reviewer_yt5X · 2023-07-16

**Confidence:** 4
**Originality:** Very Good
**Technical Quality:** Very Good
**Clarity Of Presentation:** Excellent
**Impact:** 3

**Recommendation:**

Weak Accept: I recommend accepting the paper, but will not argue for my recommendation if the majority of other reviewers have a different opinion.

**Review:**

In my opinion, this is a strong paper. I truly did not find a lot to criticize here. The problem being addressed is pertinent, the proposed solutions makes sense, the results are thorough, and the writing is clear.

One limitation I would like to point out which is not mentioned in the paper is the requirement of a good realistic closed-loop simulator. Most existing simulators are either not closed-loop or the closed-loop behavior of other agents is unrealistic (such as nuPlan). Applying this approach on actual vehicle testing is not scalable. It is not practical to generate a scenario with some specific lane geometry and number of agents, etc., and then go out in the real-world to seek it. So from that perspective, this approach and others reliant on the development of good closed-loop AV simulators will not have a very significant impact in the near future.

Another point is that safety is viewed as merely collisions in this paper, measured with the distance from other agents. Establishing safety encompasses a lot more than that. Reasoning about the impact on AV's actions on other agents' behavior is also crucial. For instance, if the AV runs through a traffic light and manages to avoid any collisions because of the quick reaction of other road agents, this behavior should still be deemed unsafe even though no collision occurs. This is less of a gripe with the paper's core methodology, more with the safety metric used.

----------- Post-rebuttal -----------

The key contribution of this paper is an approach to achieve improved "targeted" coverage of the ODD, which the paper demonstrates fairly well. I still view this paper positively and believe that it should be published. However, it seems that I share the same concern as other reviewers regarding the limited impact of the paper due to the need for good closed-loop simulators (as was raised in my initial review). So, to ensure that my score is aligned with the content of my comments, I have updated my score to Weak Accept.

**Quality Of The Limitations Section:**

Limitations are addressed clearly

**Questions For Rebuttal:**

Please include more details of the simulator used in the tests. What simulator was used? How are the road agent reactions generated?

**Robotics Focus:**

Highly relevant to robotics but no hardware experiments

**Summary Of Paper:**

Autonomous vehicle (AV) driving scenarios are uncountable and depend on a huge number of factors, such as agents' states, intents, lane geometry, number and type of agents, weather, etc. Making a safety case for the AVs requires establishing an operational design domain (ODD) and proving safety over all scenarios that arise within this ODD, i.e., _coverage_.

As one can anticipate, the challenge to this task lies in testing these immense number of combinations. This paper provides a method for coverage testing by probabilistically modeling the safety outcomes of scenarios using Gaussian Process (GP). Scenarios that satisfy the safety threshold with high probability are deemed safe, while those that fail with high probability are deemed unsafe. The uncertain scenarios are slotted for further testing. This enables more scalable coverage testing by focusing on scenarios where the safety outcome is uncertain, instead of testing every possible combination.

**Summary Of Recommendation:**

I recommend strong accept. There are a couple of issues that I have listed in my review above, but they are minor. Overall, the method presented in this paper is sensible and it outperforms the existing methods.

----------- Post-rebuttal -----------

I have updated my score to Weak Accept due to the potentially limited impact of the paper.

---

### Official Review · Reviewer_YzeP · 2023-07-19

**Confidence:** 4
**Originality:** Good
**Technical Quality:** Fair
**Clarity Of Presentation:** Fair
**Impact:** 2

**Recommendation:**

Weak Reject: I recommend rejecting the paper, but will not argue for my recommendation if the majority of other reviewers have a different opinion.

**Review:**

Quality:
- The State of the Art of the paper is quite poor and does not cover the extensive work and papers that were published in the field of scenario based testing or virtual testing for AVs. It is recommend to further read similar papers in order to create a clear contribution for the own paper.
- Currently, some statements are wrongly made in the paper e.g. "Intuitively, if the AV easily passed a particular test, it can
53 pass similar tests with high probability." -> This is not true since specific scenario parameters can lead to catastrophic AV behavior e.g. changing the weather in the same scenario and same paramter setup


Clarity:
- The paper lacks of clarity and is confusing many times with the goals, objectives and content of the paper
- The authors use the terms "testing" and "Testing framework" similarly, unfortunately there are clear differences between the individual terms and the topcis. The authors need to state clearly what they are doing e.g. right now its a scenario generation framework. There is no new simulation or similarly invented.
- Currently i am missing a clear definition from a functional safety (FuSa - ISO 26262) and Safety Of The Intended Functionality (SOTIF -ISO/PAS 21448) point of view how the approach can be applied and how it is justified. Right now existing approaches need to go over certain scenarios and parameter sets in order to cover everything to be correct from a SOTIF and FuSa perspective. Yes, the author cited both ISOs but miss where the approach does fit into those.

Originality:
Applying the method of Gaussian process in order to leverage observed test results to model the probability of passing or
failing across the entire parameter space is a neat approach. I think the authors demonstrate on their toy problem and simple "Pass-uncertain-fail" setup that the approach can be quite useful


Significance:
While the paper exhibits promising problem formulation and methodological approaches, its lack of comprehensive evaluation, results, and discussion, particularly in the context of Robotics and Autonomous Vehicles (AV), diminishes its significance for the field, highlighting the missed opportunity to showcase the potential and contribution of the research. Additionally, the absence of a demonstration of the paper's integration with other simulation and testing environments, such as CommonRoad, further limits its relevance and applicability within the broader research landscape.


Strength:
-  The paper addresses a crucial and timely challenge in the field of autonomous vehicles - understanding system safety conditions. With the rapid advancement of AV technology, the need for robust testing frameworks becomes increasingly critical, making this research highly relevant.
- The paper introduces a novel problem formulation to partition the scenario parameter space without relying on discretization. By using a probability-based approach, the authors avoid the limitations of traditional binning methods, enabling a more accurate representation of safe, unsafe, and unknown regions.


Weaknesses:
- The paper primarily focuses on simulation-based testing, but there is limited discussion or validation of the framework's effectiveness in real-world scenarios. Without empirical validation in real-world environments, the applicability and reliability of GUARD might remain uncertain.
- the paper briefly introduces the use of Gaussian Processes for probability modeling, but it lacks a detailed explanation of the underlying assumptions and limitations of this approach. A more comprehensive discussion would help readers understand the reliability of the probability estimates. Additionally it is recommend to enhance the limitation and discussion section to show the reader where and why the approach can create significance and impact.
-The paper suffers from unclear simulation experiments and results, which significantly hinders its overall clarity and impact. The lack of a well-defined and transparent description of the simulation setup and execution raises questions about the validity and reproducibility of the findings.
- Additionally, the presentation of the results is inadequate, with insufficient data visualization or statistical analysis to support the authors' claims effectively. As a result, readers may find it challenging to interpret or draw meaningful conclusions from the simulation outcomes


**Quality Of The Limitations Section:**

Limitations are not well addressed

**Questions For Rebuttal:**

- Although the paper describes the theoretical foundations of GUARD, it does not provide sufficient implementation details or practical guidelines, potentially making it challenging for practitioners to replicate and apply the framework effectively. Right now the paper is missing the context of FuSa, SOTIF and the connection to other simulation frameworks. Where do the authors see their method?
- Right now, the Generalizability to Various AV Systems is quite unclear. The paper lacks an exploration of how the GUARD framework may adapt to different types of autonomous vehicle systems or the challenges that could arise in diverse AV architectures. This focuses especially on different hardware and software architecture setups as wells specific ODDs

**Robotics Focus:**

Relevant but unlikely to deploy to hardware in near future

**Summary Of Paper:**

The authors are tackling a problem in the field of autonomous driving or autonomous cars that drive on open streets. A problem for autonomous cars is the testing and verification of the system in order to prove the reliability and quality of the AVs software and hardware. The authors are describing the problem of virtual testing which is necessary to evaluate and verify AVs. The authors of the paper are proposing a "new" testing framework called GUARD which relies on Gaussian Process to generate tests for the AV. Everything displayed in the paper is done in simulation only.

**Summary Of Recommendation:**

In summary, the paper is adressing the critical problem of virtual testing for autonomous vehicles. By proposing to apply the method of Gaussian processes to generate a new testing framework/ new test scenarios in simulation, the authors want to display that their framework can evaluate more efficiently  and accurately. Currently the simulation results and statistical results do not display this claim thoroughly since the result and discussion section is missing critical content and explanations.

---

### Official Review · Reviewer_WNoS · 2023-07-22

**Confidence:** 4
**Originality:** Very Good
**Technical Quality:** Very Good
**Clarity Of Presentation:** Good
**Impact:** 4

**Recommendation:**

Weak Accept: I recommend accepting the paper, but will not argue for my recommendation if the majority of other reviewers have a different opinion.

**Review:**

= Contribution

The algorithm proposed by the author is sound and shown to outperform several baselines. The paper is generally clear, though one weakness lies in the way the research idea is communicated. It seems to this reviewer that instead of focusing on the problem setup and experimental context, the authors should have simply addressed the problem in a more algorithmic way. This would have made the contribution much more clear.

= Research Idea

== Quality

The paper is very well executed. The contribution is well-situated and argued. The algorithmic solution is quite appropriate, it seems that this kind of algorithm should already exist in another context though. The idea is very similar to Bayesian optimization, with the key difference that the goal is to characterize a hypersurface. The paper presents a series of comparative experiments highlighting the benefit of the approach.

== clarity

The paper is quite clear, though the focus is somewhat not ideally placed. The problem the authors are solving is algorithmic: How to sequence a discrete test campaign to best characterize the system binary performance on the benchmark? As such it would make the presentation much clearer to have the algorithm in the paper instead of in supplementary material. To also make the connection to Bayesian optimization stronger and to try to find in the literature, algorithms like this one applied in different contexts. Somehow the scope of the contribution should be reviewed to clarify the paper.

== originality

The paper is rather original, and it is surprising that this idea was not investigated in prior work.

== significance

This is quite significant and as said surprising that this was not investigated before. However a few limitations underly the approach. These are pointed out by the authors themselves.

= Strength and Weakness

Strengths
-	Very well-executed research idea
-	Breadth baseline comparison

Weakness
-	Does not focus with enough clarity on the algorithm underlying the method
-	The algorithm is outlined throughout the paper but only made explicit in supplementary material
-      The algorithm in the supplementary material lacks comments and is not easy to follow
-	Many hyperparameters are given but it is not clear why they are important or chosen
-	Connection to similar work is missing


**Quality Of The Limitations Section:**

Limitations are addressed clearly

**Questions For Rebuttal:**

- Could you address the focus of the paper?
- Can you do a literature review on these kinds of algorithms?

**Robotics Focus:**

Highly relevant to robotics but no hardware experiments

**Summary Of Paper:**

This paper presents a framework for finding optimal test sequences such as found in testing autonomous driving components. For instance, AV are tested on parameterizable test scenarios. The dimensions of the search space for a given scenario are the number of parameters that the scenario has. Effectively the paper proposes an algorithm for efficiently characterizing the failing manifold of scenarios using a discrete amount of probes.

**Summary Of Recommendation:**

I would really rework the presentation of the paper.
I think this paper has high impact potential but it's just not presented this way.

---

### Author Response · Authors · 2023-08-11
**General Response**

We thank the reviewers for their thoughtful reviews and valuable feedback. We are excited that all reviewers believed we investigated an important and crucial problem. And we appreciate that most found our proposed solution appropriate or interesting[*Reviewer WNoS, Reviewer  yt5X, Reviewer QJej*]. We summarize the major points of the rebuttal here and elaborate on other concerns or questions in the individual rebuttals.

**Additional details on our simulator [Reviewer YzeP, Reviewer yt5X, Reviewer QJej]**  \
We used our closed-loop simulator as it’s more realistic than existing public simulators. Our simulator follows [1] to simulate high-fidelity LiDAR sensory input and is based on commercial industry-standard LiDAR sensors. The actor assets are taken from an asset bank of high-quality real world vehicle meshes.

To evaluate our algorithm, we used scenarios designed by expert test engineers and based on guidelines[2] prescribed by the National Highway Traffic Safety Association (NHTSA). The scenarios employ a combination of reactive actors and scripted actors. The scripted actors follow a fixed parameterized trajectory to execute specific stress testing maneuvers such as cutting in front of the AV. The reactive actors act as realistic background traffic and they are modelled using the  Intelligent Driver Model[3]. The specific parameterization of the scenarios can be found in our supplementary materials, and we have revised the main paper to include these additional details.

**Algorithmic focus of paper [WNoS]**  \
The focus of the paper is to develop an efficient, accurate, and scalable coverage-based testing method. We formulate the coverage-based testing problem as one of sequencing a finite number of test scenarios to estimate the probability of passing or failing across the entire scenario parameter space. Then, under this formulation we adopt GPs, levelset sampling, learnable kernels, and propose evaluation metrics.

We have revised our writing in the introduction (section 1) to and method (section 3) to more clearly state our formulation and algorithm. In addition we have also made the algorithm box more clear and moved it to the main paper per the reviewer’s suggestion. We hope this makes the writing more clear and we are happy to incorporate additional feedback.

**Connection to functional safety methodologies [YzeP]**  \
Our testing framework is meant to be used in conjunction with functional safety assessments. First, our approach is aligned with principles introduced by ISO26262 and SOTIF. For example, evaluating if the system passes or fails with respect to requirements and aiming to understand known passes, known failures, and unknowns. When it comes to evaluating the safety of AVs in practice, functional safety experts define a comprehensive set of safety requirements for the AV, which need to be verified against all the possible scenarios from a particular ODD. However, proving the intended function under all possible scenarios in the ODD is difficult, especially if these scenarios have to be manually generated. This is where GUARD can be applied to automatically generate tests and validate that the AV behaves as intended with respect to continuous measures of the safety requirement on parameterized scenarios from the ODD.

Per the reviewer’s suggestion, we have revised our conclusion section to touch on the connection to functional safety requirements and how GUARD can be applied in practice from this perspective.

[1] https://arxiv.org/pdf/2006.09348.pdf \
[2] https://www.nhtsa.gov/sites/nhtsa.gov/files/documents/13882-automateddrivingsystems_092618_v1a_tag.pdf \
[3] https://en.wikipedia.org/wiki/Intelligent_driver_model

---

### Decision · Program_Chairs · 2023-08-30

**Decision:**

Accept (Poster)

**Comment:**

The authors' response has addressed many of the reviewers' concerns, and the general assessment is that this paper provides a good contribution to CoRL on a very timely topic, namely, testing of autonomous vehicles. I agree with this assessment and therefore recommend acceptance. That said, the reviewers had several suggestions for improvement -- in the final version of this paper, the authors should *carefully* revise the paper to address all reviewers' comments / suggestions.